# COVID-19 Vaccines Adverse Reactions Reported to the Pharmacovigilance Unit of Beira Interior in Portugal

**DOI:** 10.3390/jcm11195591

**Published:** 2022-09-23

**Authors:** Carina Amaro, Cristina Monteiro, Ana Paula Duarte

**Affiliations:** 1Health Science Faculty, University of Beira Interior, 6201-001 Covilhã, Portugal; 2UFBI-Pharmacovigilance Unit of Beira Interior, University of Beira Interior, 6201-001 Covilhã, Portugal; 3CICS-UBI-Health Sciences Research Centre, University of Beira Interior, 6201-001 Covilhã, Portugal

**Keywords:** adverse drug reactions, COVID-19 vaccines, mRNA vaccines, vaccines with a viral vector, pharmacovigilance, immunization, safety

## Abstract

Coronavirus disease 2019 is an acute respiratory disease caused by the severe acute respiratory syndrome coronavirus 2. As the virus spreads rapidly, it has become a major public health emergency, which has led to rapid vaccines development. However, vaccines can present harmful and unintended responses, which must be notified to the National Pharmacovigilance System. The aim of this study is to characterize the adverse drug reactions (ADRs) of these vaccines notified in the region covered by the Regional Pharmacovigilance Unit (RPU) of Beira Interior, in Portugal, between 1 and 31 December 2020. During this period, 4 vaccines were administered: Comirnaty^®^, Spikevax^®^, Vaxzevria^®^ and Jcovden^®^. The RPU of Beira Interior received 2134 notifications corresponding to 5685 ADRs, of which 20.34% (n = 434) of the notifications were considered serious reactions. Of these, 9.52% (n = 42) resulted in hospitalization and 0.45% (n = 2) resulted in death. Among the ADRs notified, reactions at or around the injection site, myalgia, headaches and pyrexia were the most commonly notified. Most ADRs were resolved within a few hours or days without sequelae. These ADRs are in accordance with clinical trials, the summary of product characteristics (SmPC) of each vaccine and ADR notifications from other countries. However, further studies are needed to confirm these results.

## 1. Introduction

Coronavirus Disease 2019 (COVID-19) is an acute respiratory disease caused by the severe acute respiratory syndrome coronavirus 2 (SARS-CoV-2), which first emerged in Wuhan in December 2019. Its transmission occurs by droplets, respiratory secretions and direct contact [1,2].

As the virus has a rapid spread, it has become a serious public health emergency [2]. Given that vaccination can be used to prevent infections or reduce the seriousness of a disease, some strategies were studied to generate vaccines against the new coronavirus, including vaccines based on DNA and RNA [3,4].

Nucleic acid vaccines consist of mRNA with information against coronavirus-specific structural proteins and do not contain any viral proteins capable of causing disease. The mRNA is taken up by cells and translated into a viral antigen, the spike protein. When recognized by the immune system as something foreign, antibodies are produced, and T cells are activated to attack the protein. If the vaccinated person later comes into contact with coronavirus, their immune system will recognize the spike protein and be ready to defend itself [5,6,7,8,9]. In the European Union (EU), during the study period, 2 mRNA-based vaccines were authorized by the European Medicines Agency (EMA): Comirnaty^®^ and Spikevax^®^ [7,8,9].

In addition to mRNA vaccines, there is another type of vaccine approved for immunization against COVID-19: vaccines with a viral vector without the ability to replicate. This type of vaccine is produced from another virus (e.g., adenovirus) that has been modified to contain information regarding the virus of interest, which will be delivered to human cells. The viral vector is a harmless virus and different from coronavirus, so it does not cause the disease. It enters human cells and releases the gene that encodes the spike protein present in SARS-CoV-2. It then uses the cell’s machinery to produce this glycoprotein which, when recognized by the immune system, leads to the production of antibodies and activation of T cells, as is in nucleic acid vaccines [10,11,12,13]. In the EU, during the study period, two vaccines based on viral vectors were authorized by the EMA: Vaxzevria^®^ and Jcovden^®^ [7,12,13].

Although medicines are essential elements in the treatment of pathologies, diagnosis and prevention, they also have risks. Thus, due to the fact that there is a limited knowledge of the therapeutic profile of some drugs, it is important to continue to monitor their safety after marketing, through several available methodologies, one of which is the notification of adverse drug reactions (ADRs) to the National Systems of Pharmacovigilance (NPS), present around the world, and created in order to monitor the safety of medicines. The NPS cover all information related to ADRs and guarantee the safety of users who have contact with medicines, especially medicines recently introduced on the market. This is the case of vaccines used to immunize against COVID-19 [14,15].

In 1992, the SNF was created in Portugal, currently coordinated by INFARMED, I.P. In the early 2000s, the SNF was decentralized into 4 Regional Pharmacovigilance Units (RPU)-Norte, Centro, Sul and Açores-, with the aim of publicizing the system and promoting notification, bringing the system closer to health professionals and promote the involvement of university centres. Since 2017, the number of RPUs in the SNF increased, and there are currently 10 covering different areas in the country. The RPU of Beira Interior is placed at the University of Beira Interior, in the interior of Portugal, covers the districts of Castelo Branco, Viseu and Guarda and involves some under reporting [16,17].

When the appearance of ADRs is suspected, the process of spontaneous notification through an online or paper form or by telephone becomes important. Spontaneous reporting is a voluntary pharmacovigilance methodology, which consists of reporting an ADR associated with a particular drug and a patient, which can be by the patient, a family member or a healthcare professional. Spontaneous reporting makes it possible to detect ADRs that occur rarely or unexpectedly, generating an alert signal for subsequent epidemiological studies [16].

Therefore, pharmacovigilance, a science involved in the detection, evaluation and prevention of ADRs, through the methodology of spontaneous reporting is an essential step to assess the safety of vaccines used in the immunization against COVID-19 [17,18,19,20].

Thus, this study had two objectives. The first objective was to characterize the ADRs associated with vaccines used in the immunization against COVID-19, notified in the region covered by the Regional Pharmacovigilance Unit of Beira Interior, in Portugal. The second objective was to compare the results obtained with the safety data from studies carried out in other countries around the world. This period was the subject of study since it was the initial period in which the vaccines authorized in the EU began to be administered to the Portuguese population.

## 2. Materials and Methods

This work is a retrospective observational study. The data under analysis were collected through spontaneous notifications sent to the Portuguese NPS by healthcare professionals, patients or family member. The Portuguese database is “Portal RAM” which is coordinated by the National Authority of Medicines and Health Products, I.P. (INFARMED). The search was carried out in this database taking into account the International Common Denomination of each vaccine (Comirnaty^®^, Vaxzevria^®^, Spikevax^®^ and Jcovden^®^), study period (1–31 December 2021) and the area covered by the RPU of Beira Interior, in Portugal.

Statistical analysis of the data obtained was performed using the Microsoft Office Excel 365 tool. In this tool, the data were organized according to the variables under study and were later represented in tables and appropriate graphics.

It is important to note that each notification concerns a single patient. However, more than one ADR and seriousness criteria may be associated with each notification. Of the 2145 notifications received, only 2134 were studied because of the lack of information in 11 notifications.

It is also important to mention that this work did not require prior authorization from the Ethics Committee, since the patients’ personal information was not used.

### 2.1. Population

The study population comprised only cases of suspected ADR associated with vaccines used in the immunization against COVID-19, notified to the Regional Pharmacovigilance Unit of Beira Interior, and no age restriction was imposed.

### 2.2. Variables

#### 2.2.1. Characterization of the Notification Source

##### Notifier Characterization

ADRs can be notified by professionals in the pharmaceutical industry, patients, family members or healthcare professionals. The healthcare professionals considered are classified as physicians, pharmacists, nurses or other healthcare professionals. These professionals play a crucial role in reporting ADRs, with the aim of reducing the negative outcomes associated with them. Bearing in mind that this study only focuses on a regional unit, there are no notifications from the pharmaceutical industry, since these professionals notify directly to the RAM portal, and no specific region is assigned to them.

##### District of Origin

For the study, only the RPU of Beira Interior was considered, which covers 3 districts: Castelo Branco, Guarda and Viseu. This area corresponds to about 700,000 inhabitants and about 8000 healthcare professionals (doctors, nurses and pharmacists). Thus, the notifications were analyzed according to the district of origin.

#### 2.2.2. Demographic Characterization of the Population

The analysis was carried out by characterizing the notifications by age and gender. In terms of age, 8 age groups were considered: 5 to 11 years old; 12 to 17 years old; 18 to 24 years old; 25 to 49 years old; 50 to 64 years old; 65 to 79 years old; ≥80 years; and unknown. Gender was classified as male, female or unknown.

#### 2.2.3. Characterization of Adverse Drug Reactionss

##### Characterization by Administered Vaccine

Notified ADRs were classified based on the associated vaccine. Only 4 vaccines were considered: Comirnaty, Spikevax, Vaxzevria, and Janssen as they were the vaccines authorized for administration in Portugal during the study period.

##### Analysis of ADRs

The description of each ADR is performed by the notifier, which is later coded according to the Medical Dictionary for Regulatory Activities (MedDRA) terminology. MedDRA is an international medical terminology developed in 1994 by the International Conference on Harmonization (ICH). Prior to the creation of this dictionary, there was no international medical terminology, so the existence of multiple terminologies created several problems in the analysis of data related to pharmaceutical products. In this way, there was a need to create an international medical terminology, in order to facilitate communication between the various health professionals, and the crossing of data regarding pharmaceutical products [21].

MedDRA terms are hierarchically organized into: system organ class (SOC); high level group term (HLGT); high level terms (HLT); preferred term (PT) and lowest level term (LLT), with the SOC level being the widest and most comprehensive and the LLT level being the most specific [21].

Thus, the ADRs were initially grouped according to the SOC group, and finally, they were organized according to the PT term.

##### Description in the Summary of Drug Characteristics (SmPC)

In order to verify the previous descriptions of the ADRs under study associated with the vaccines, the SmPC of each vaccine was used, grouping the data into 2 categories: “Described in the SmPC” and “Not described in the SmPC”.

Regarding ADRs not described, they were grouped into 2 parameters: “Degree of Causality Studied” and “Degree of Causality not studied”. Causality is attributed by an expert from the regulatory authority or the pharmaceutical company, in ADRs considered serious, and from the information provided during the notification.

Subsequently, the ADRs for which the degree of causality was studied were grouped into 6 categories: Definitive, Probable, Possible, Unlikely, Conditional, Unclassifiable [22].

##### Seriousness and Seriousness Criteria

Regarding the seriousness, notifications were grouped into serious and non-serious based on the notifier’s assessment and/or Regional Pharmacovigilance. Subsequently, serious ADRs were grouped by seriousness criteria.

There was also the characterization of ADRs associated with the seriousness criteria “Hospitalization”, “Life Risk” and “Death”, according to age and associated vaccine brand.

An ADR is considered serious if it “results in temporary or permanent disability, causes a congenital abnormality, results in hospitalization or prolongation of hospitalization, causes death or is life-threatening, or fulfills another clinically important condition” [23].

##### Evolution of ADRs

Data were grouped into the following categories: Cure, Cure with collateral damage, in recovery, Death and Unknown, based on the information provided by the notifier.

##### Characterization of Notifications with the Outcome “Death” with Terms Belonging to the IME List

Finally, the characterization of the notifications that culminated in death was carried out, taking into account the presence of terms belonging to the Important Medical Events (IME) list.

In order to facilitate the classification of ADRs, as well as assist in the analysis of notifications submitted to the National Pharmacovigilance Systems, Eudravigilance created a list of medical terms considered important, called the IME list, based on the definitions adopted by the ICH. Important Medical Events are events that may not immediately lead to death or hospitalization but compromise the individual’s life or require medical or surgical intervention in order to avoid the outcomes listed in the definition of seriousness ADR [24].

## 3. Results

As mentioned above, the number of notifications to be analyzed, after the duplicate and annulled notifications have been withdrawn, was 2134.

### 3.1. Characterization of the Notification Source

#### 3.1.1. Notifier Characterization

In this study, the type of notifier who submitted the notification was analyzed. Through Figure 1, it is possible to observe that most notifications were submitted by the pharmacist (82.15%, n = 1753), followed by the user or other non-healthcare professional (6.47%, n = 138) and later the physician (6.09%, n = 130). Nurses had a notification rate of 4.87% (n = 104), and finally, other healthcare professionals submitted only 0.42% (n = 9) of the notifications.

#### 3.1.2. District of Origin

For the study, only the RPU of Beira Interior was considered. Through Figure 2, it is possible to observe that most notifications presented Castelo Branco as the district of origin (86.36%, n = 1843), followed by Viseu (9.18%, n = 196) and finally Guarda (4.45%, n = 95).

### 3.2. Demographic Characterization of the Population

The notifications were characterized as to the age and gender of the patients, as can be seen in Table 1 and Figure 3, respectively.

Analyzing Table 1, it is possible to verify that most notifications were associated with patients aged between 25 and 49 years (57.64%, n = 1230), followed by patients aged between 50 and 64 years. (26.90%, n = 574).

Regarding gender, notifications were grouped into female, male and unknown. The female gender presented the most notifications, accounting for 1534 (71.88%) out of 2134 total notifications.

### 3.3. Characterization of Adverse Drug Reactions

#### 3.3.1. Characterization by Administered Vaccine

During the study period, only 2 types of vaccines were administered–mRNA vaccines and non-replicating viral vector vaccines. It is important to note that each notification concerns a single vaccine. According to Figure 4, mRNA vaccines were highlighted (82.52%, n = 1761). In turn, vaccines with a non-replicating viral vector had a notification rate of 17.48% (n = 373).

Subsequently, the ADRs were organized into 4 classes, according to the brand name of the administered vaccine. Analyzing Figure 5, it is possible to observe that most of the notifications submitted were associated with the Comirnaty vaccine (79.29%, n = 1692), followed by the Vaxzevria vaccine with 325 notifications (15.23%). The Spikevax vaccine had a notification rate of 3.23% (n = 69), and finally, the Jcovden vaccine was associated with 2.25% (n = 48) of the notifications submitted.

#### 3.3.2. Analysis of Adverse Drug Reactions

In this study, the notifications sent to the Regional Pharmacovigilance Unit of Beira Interior were initially characterized according to the SOC classification of the MedDRA dictionary. The 2134 notifications were organized into 5685 SOC reactions, meaning that there were, on average, approximately 3 SOC reactions for each notification.

Through Table 2, it is possible to conclude that the three SOC groups most frequently notified were “General disorders and administration site conditions”, “Nervous system disorders”, and “Musculoskeletal and connective tissue disorders”, presenting the following frequencies 2454 (43.17%), 1048 (18.43%) and 1015 (17.85%), respectively. The least notified SOC groups were “Pregnancy, puerperium and perinatal conditions”, constituting 0.02% (n = 1) of the notifications, and “Social Circumstances”, with 0.04% (n = 2).

After the analysis by SOC groups, the ADRs notified to the URF of Beira Interior were classified according to the PT terms. The results are shown in Table 3. The “Other reactions” category encompassed adverse reactions with a notification rate ≤1%.

Through Table 3, it is possible to conclude that the 3 ADRs most frequently notified were “Reaction at or around the site of administration”, “Myalgias”, and “Headache”, with the following frequencies 1147 (20.18%), 751 (13.21%) and 608 (10.69%), respectively.

Regarding the “Other reactions” category, it included ADRs with a low reporting rate, such as miscarriage, anaphylactic shock, seizures, respiratory distress and syncope.

#### 3.3.3. Description in the Summary of Drug Characteristics (SmPC)

The ADRs were analyzed in terms of their prior knowledge. Thus, the SmPC of the respective vaccines under study was used for further characterization in: “Described in the SmPC” and “Not described in the SmPC” [25,26,27,28].

Through Figure 6, it is possible to verify that 5299 ADRs were described in the SmPC (93.21%), among which it is possible to highlight “Reaction at or around the administration site”, “Myalgias”, “Arthralgias”, “Pyrexia” and “Headaches”. 386 ADRs were not described in the SmPC (6.79%).

ADRs not described in the SmPC were grouped according to 2 categories: “Degree of Causality Studied” and “Degree of Causality not studied”. The category of “Degree of Causality Studied” obtained the most prominence (61.66%; n = 238) (Figure 7).

The ADRs for which the degree of causality was studied were grouped into 6 categories: “Definitive”, “Probable”, “Possible”, “Unlikely”, “Conditional”, and “Unclassifiable”. According to Table 4, it is possible to observe that most ADRs were classified as “Possible” (n = 118, 49.58%). 100 ADRs (42.02%) have a “Probable” degree of causality and 18 (7.56%) were classified as “Unlikely”. The degree of causality that was less prominent was the “Unclassifiable” (n = 2; 0.84%). The “Definite” and “Conditional” degrees of causality were not assigned to ADRs notified.

#### 3.3.4. Seriousness and Seriousness Criteria

##### Seriousness

In this study, the 2134 notifications were further characterized according to seriousness. Among these notifications, 1700 notifications were considered non-serious (79.66%) as they did not fulfill any of the criteria mentioned in Seriousness and Seriousness Criteria section. In turn, 434 notifications were considered serious, representing 20.34%, as can be seen in Figure 8.

Subsequently, the 434 notifications considered serious were organized according to the brand of vaccine administered. Figure 9 shows that most ADRs considered serious belonged to the Comirnaty vaccine (70.51%, n = 306), followed by the Vaxzevria vaccine with 60 serious ADRs (13.82%). The Jcovden vaccine had 35 serious notifications (8.06%) followed by Spikevax with 33 notifications (7.60%).

##### Seriousness Criteria

ADRs considered severe were grouped by seriousness into 5 criteria: “Clinically important”, “Disability”, “Hospitalization”, “Life Risk” and “Death”. Some serious notifications had more than one seriousness criteria, with a total of 441 seriousness criteria notified for 434 serious notifications.

The seriousness criteria “Clinically important” had great prominence, with a percentage of 64.63% (n = 285), followed by “Disability” (24.04%, n = 106). Then came the criteria “Hospitalization” with 9.52% of serious notifications, which corresponds to 42 notifications. Finally, the criteria “Life Risk” and “Death” appeared with 6 and 2 notifications (1.36% and 0.45%), respectively, as can be seen in Figure 10.

Then, the ADRs with seriousness criteria “Hospitalization”, “Life Risk” and “Death”, were characterized according to patient’s age and associated vaccine brand.

Initially, there was a characterization of the 42 notifications with the seriousness criteria “Hospitalization” according to the brand of vaccine administered and age, and the results can be found in Figure 11 and Table 5, respectively. Regarding the brand of the associated vaccine, the Comirnaty vaccine was responsible for most hospitalizations (45.24%, n = 19), followed by Vaxzevria responsible for 35.71% (n = 15). The Jcovden vaccine was associated with 14.29% (n = 6) of notifications with the seriousness criteria Hospitalization, and finally, Spikevax originated 4.76% (n = 2) of hospitalizations.

Analyzing Table 6, most hospitalizations were associated with patients aged between 25 and 49 years (30.95%, n = 13), followed by patients aged between 65 and 79 years (28.57%, n = 12).

Subsequently, the 6 notifications with the seriousness criteria “Life Risk” were characterized, and the results are found in Figure 12 and Table 7. In Figure 12 it is possible to observe that the vaccines responsible for this seriousness criteria were the Comirnaty vaccine, the Spikevax vaccine and the Jcovden vaccine, each responsible for three, two and one case, respectively. In fact, the most prominent vaccine was Comirnaty, the most administered vaccine in Portugal [29].

Regarding the age groups associated with the seriousness criteria “Life Risk”, it’s possible to observe through the Table 6 that the age group with the most cases was the group from 25 to 49 years old (50.00%, n = 3). The age groups from 18 to 24 years old, 50 to 64 years old and ≥80 years old were associated with a single case of “Life Risk”.

Finally, the 2 notifications associated with the seriousness criteria “Death” were characterized. In Figure 13 it’s possible to observe that the vaccines responsible for this seriousness criteria were the Comirnaty and the Vaxzevria vaccine, each responsible for one case. These patients were 76 and 84 years old, respectively.

#### 3.3.5. Evolution of Adverse Drug Reactions

Another important parameter to be evaluated is the evolution of the patient’s clinical condition, whose data allow us to understand the possible risks to the patient’s life after the administration of the drug in humans (Table 7).

Of the 5685 ADRs notified, 5361 (94.30%) resolved within a few hours or days without the appearance of collateral damage, 9 (0.16%) led to the appearance of collateral damage, 6 (0.11%) ADRs patients were in recovery at the time of reporting and in 6 (0.11%) ADRs the result was death. In 5.33% of the ADRs (n = 303), it was not possible to obtain information regarding the outcome of the reaction.

#### 3.3.6. Characterization of Notifications with the Outcome “Death” with Terms Belonging to the Important Medical Events List

In total, 2 notifications were obtained that culminated in death, corresponding to a total of 6 ADRs, of which 5 were on the IME list. Through Table 8, it is possible to see that the 6 ADRs associated with the outcome “Death” belonged to 3 SOC groups, among which the “Cardiac disorders” group was more prominent.

These 2 cases obtained a causality study by the regulatory authority, with a conclusion of an unlikely causal relationship, i.e., there was no causal relationship between ADR and the associated vaccine.

## 4. Discussion

This study allowed the characterization of the notifications of ADRs associated with vaccines used in the immunization against COVID-19, notified to the Pharmacovigilance Unit of Beira Interior, in Portugal, in the time period between December 2020 and December 2021.

Initially, the type of notifier who submitted the notification was analyzed. Through data analysis, it was found that pharmacists had the highest notification rate. The user or other non-healthcare professional also contributed to improving the safety profile of vaccines, followed later by physicians. According to the graph “Evolution of ADR Notifications received in the SNF, by origin, 1992–2021”, made available by INFARMED, over the years, the biggest notifier of ADRs has been the pharmaceutical industry [30]. In the year 2021, the industry was the biggest notifier, followed by physicians and later by pharmacists. The results obtained in this study are not in accordance with the INFARMED graph, presenting the pharmacist as the greatest notifier of ADRs associated with vaccines used in the immunization against COVID-19. This is related to the fact that the pharmaceutical services of 2 hospital centers established pharmacovigilance protocols and were later involved in collecting information from patients after the administration of these vaccines, thus increasing the notification rate obtained by pharmacists. However, in general, the results obtained in the study support the fact that healthcare professionals are increasingly aware of the need to notify suspected ADRs, in order to improve the safety profile of medicines. As mentioned in the Notifier Characterization section, the fact that this study focused only on a regional unit explains why we did not obtain ADRs notified by the pharmaceutical industry, as these professionals notify them directly on the ADR portal, with no specific region being assigned to them.

Regarding the district of origin of the notification, it was found that most notifications presented Castelo Branco as the district of origin, followed by Viseu and finally Guarda.

Among the analyzed data, the age group with the highest notification rate, among the groups considered, was the group from 25 to 49 years old, followed by patients aged between 50 and 64 years old. These data are in accordance with the document “Pharmacovigilance Report: Monitoring the safety of vaccines against COVID-19 in Portugal. Data received until 01/31/2022.”, which indicates that the age groups 25 to 49 years old and 50 to 64 years old had the highest number of ADR notifications at the national level [29]. This fact may be due to the greater number of vaccines being administered in these two age groups, in addition to being age groups with greater ability to recognize an ADR. It is also important to point out that, according to the data referring to the resident population in Portugal by age group, these are the age groups with the largest population [31]. Regarding gender, the population was mostly female, and these data are supported by the aforementioned document [29]. Generally, the female gender is the one that most notified to any drug, due to the greater susceptibility to develop ADRs, compared to the male gender, probably due to the physiological differences between both sexes. Additionally, it may be related to women’s greater attention to their health and signals developed by their body [29,32,33]. International studies also confirm the aforementioned data regarding the age groups and gender with the highest notification rates of vaccine-associated ADRs [34,35].

Subsequently, ADRs were analyzed according to the type and brand of the vaccine administered. In this study, during the analysis period, mRNA vaccines were the most notified. Most of the notifications submitted were associated with the Comirnaty vaccine, followed by the Vaxzevria vaccine. In fact, these data were in agreement with the INFARMED Pharmacovigilance Report, which indicates that the majority of ADRs notified in Portugal correspond to mRNA vaccines [29]. It should be noted that these results may be due to the fact that these were the most administered vaccines in Portugal as well as in the EU and the United States [36,37]. The same report mentions that the vaccine with the highest number of ADR notifications was Comirnaty, followed by the Vaxzevria vaccine, which reinforces the data in Figure 5 available in Section 3.3.1. of the results [29]. These data are corroborated by EMA data, which indicate that these were also the most ADR vaccines reported in Europe [38]. According to data from the Centers for Disease Control and Prevention, the Comirnaty vaccine was the vaccine with the most ADRs reported in the United States [39].

The three most frequently notified SOC groups were “General disorders and administration site conditions”, “Nervous system disorders”, and “Musculoskeletal and connective tissue disorders”. Thus, with regard to “General disorders and administration site changes”, it is easy to see why they represented the most frequent SOC group, given that this group encompassed non-specific symptoms that affect various sites in the body, such as general malaise or fatigue, as well as ADRs frequently associated with the administration of any vaccine, such as pain, swelling, itching or bruising at the injection site. These signs are usually mild and transient. As for the second group, “Nervous System Diseases”, they included symptoms such as headaches, migraines and convulsions. The third most notified group was “Musculoskeletal and connective tissue disorders”, which included myalgias, arthralgias and pain in the extremities. It is easy to understand why it was among the three most notified groups, as SARS-CoV-2 binds to host cells through the spike glycoprotein, through the ACE2 receptor [40]. This receptor is found in the epithelial cells of the pulmonary alveoli and in the enterocytes of the small intestine, as well as in the skeletal muscle and central nervous system, which may be related to myalgias. Additionally, another phenomenon associated with myalgias is the “cytokine storm”, in which interleukin-6 plays a key role in inducing the production of prostaglandin E2, associated with inflammation and pain [40]. The vaccines used to immunize against COVID-19, despite not containing the virus in their constitution, have information that encodes the spike protein, recognized by the immune system, and capable of causing an inflammatory response, through a large production of pro-inflammatory cytokines, which leads to the appearance of myalgias and other musculoskeletal symptoms [41,42]. These data are in accordance with the aforementioned Pharmacovigilance Report [29]. According to the document “Rapporto sulla Sorveglianza dei vaccini anti-COVID-19”, issued by the Agenzia italiana del farmaco, the three most frequently notified SOC groups were the “General disorders and administration site conditions”, “Nervous system disorders”, and “Musculoskeletal and connective tissue disorders” which are in accordance with the results obtained through this study [43].

Following the analysis by SOC groups, the ADRs were classified according to the PT terms. The three most frequently notified ADRs were “Reaction at or around the injection site”, “Myalgias”, and “Headache”. These data are in agreement with the data from the most frequently notified SOC groups referred to in the previous paragraph (“General disorders and administration site disorders”, “Nervous System Disorders”, and “Musculoskeletal and connective tissue disorders”). The explanation for the ADR “Reaction at or around the injection site” being among the three most frequently reported ADRs is related to the way in which the vaccines are administered. This group belongs to the SOC group “General disorders and administration site conditions”, referred in the previous paragraph. “Headache” is among the three most reported ADRs, however the mechanism by which it occurs remains unclear. Some authors suggest that “Headache” may be due to a direct activation of the trigeminal vascular system, which consists of nerve fibers originating from the trigeminal nerve that innervate cerebral blood vessels. Additionally, another phenomenon possibly associated with “Headaches” is the “cytokine storm”, associated with inflammation and pain [44]. These data are in agreement with the INFARMED Pharmacovigilance Report, which mentions “Headache”, “Myalgias” and “Pain at the injection site” among the most frequently notified ADRs. This report also mentions that “Pyrexia” is the most prominent ADR, something that was not verified through the data under study [29]. However, this is a term that also has a high notification rate, ranking 4th in Table 3 of Section 3.3.2. It is easy to understand why it had a high notification rate, taking into account that it is characterized by an immune system response to a foreign body introduced into our system, as is the case with the vaccine [39,40]. These data are also corroborated by studies carried out in other countries, which indicate that “Reaction at or around the injection site”, “Myalgias”, “Pyrexia” and “Headache” are among the ADRs most commonly associated with vaccines [45,46,47]. During the study period, several countries suspended or restricted the use of vaccines to certain populations due to the emergence of rare adverse reactions [48,49,50].

The notified ADRs were compared with the SmPCs of the respective vaccines, showing that most RAMs were already described. Regarding the degree of causality, most ADRs were classified as “Possible” or “Probable”, followed by the degree “Unlikely” and the degree “Unclassifiable”. Although ADRs not described in the SmPCs represent a low percentage, it is crucial that they receive a degree of importance, especially in those in which it was possible to conclude the degree of causality as “Possible” and “Probable”, since they allow updating the safety profile of each vaccine and consequently its SmPCs, thus reinforcing the importance of reporting ADRs.

Regarding seriousness, 20.34% of ADRs were considered serious ADRs, most of which were associated with the Comirnaty vaccine. In fact, these data are in agreement with the INFARMED Pharmacovigilance Report, which indicates that the majority of ADRs associated with vaccines used in the immunization against COVID-19 correspond to non-serious ADRs [29]. According to the document “Rapporto sulla Sorveglianza dei vaccini anti-COVID-19”, most ADRs were classified as non-serious, which corroborates the results obtained in this study. However, this document indicates that the vaccine most associated with serious reactions was Spikevax, which is not in agreement with the results obtained in this study [43]. Our results may be due to the fact that Comirnaty was the most administered vaccine in Portugal [29]. Even so, there is a large percentage of serious notifications, which again reinforces the importance of healthcare professionals and users to carry out the notifications of ADRs.

The seriousness criteria with the highest rate were the “Clinically important” criteria, followed by the seriousness criteria “Disability”, “Hospitalization”, “Life risk” and “Death”. It is important to note that there were notifications in which the notifier had selected more than one seriousness criterion. These data are in accordance with the documents “Pharmacovigilance Report: Monitoring the safety of vaccines against COVID-19 in Portugal. Data received until 01/31/2022” and “Rapporto sulla Sorveglianza dei vaccini anti-COVID-19”, which indicates that the seriousness criteria that stands out the most is “Clinically important”, with the criteria “Life Risk” and “Death” being less prominent [29,43]. The Comirnaty vaccine was responsible for the most hospitalizations as well as for half of the cases associated with the “Life Risk”. In fact, these results may be due to the fact that this was the most administered vaccine in Portugal as well as in the EU and in the United States [36,37]. Regarding the seriousness criteria “Death”, the associated vaccines were Comirnaty and Vaxzevria, with patients aged 86 and 74 years, respectively, both of whom had a history of acute myocardial infarction.

Most of the ADRs notified progressed to cure. In total, there were 2 notifications that progressed to death, of which 5 terms were on the IME list, most of the terms referred to “Cardiac disorders”. These 2 notifications corresponded to patients aged 74 and 86 years, with a history of acute myocardial infarction as well as in the presence of cardiovascular risk factors, among which diabetes mellitus, arterial hypertension, obesity and dyslipidemia stand out. Regarding the notified ADRs, the regulatory authority classified them with a degree of causality “Improbable”, based on the history that the patients had, which meant that the vaccines were not the cause of death for both patients. These data are in agreement with several studies that indicated that the cases of death that occurred in patients after vaccination against COVID-19 were not related to the vaccines administered, being nothing more than mere coincidence [51].

### 4.1. Limitations

This study had some limitations, among which we can highlight the rate of under reporting of suspected ADRs, i.e., not all ADRs that occurred were notified to the National Pharmacovigilance System, which may have occurred, for example, due to lack of time or ignorance regarding the existence of the “Portal RAM” [16,17]. Another limitation is related to the fact that some notifications presented a lack of information, making their study difficult. Additionally, the fact that the Comirnaty vaccine is the most administered vaccine in Portugal, leading to the majority of ADRs reported being associated with this vaccine, may have biased the results obtained, since the vaccines under study were not administered in the same number of patients, making it more difficult to compare the results.

### 4.2. Strengths of Study

The strengths of our study included a large sample size in which it was possible to characterize several parameters associated with the reported ADRs. Additionally, it was the first study carried out in Portugal, to our knowledge, over a long period involving data corresponding to the first, second and third doses.

## 5. Conclusions

The notification of suspected ADRs was mainly related to common and non-serious reactions (e.g., pyrexia, fatigue, myalgia and reaction at or around the injection site), which is in accordance with clinical trials, in the vaccine SmPCs and in ADR notifications of vaccines from other countries. In general, ADRs resolved within a few hours/days without any consequences, which confirms a favorable safety profile of the COVID-19 vaccines. Despite the results obtained, further studies are needed to confirm these data.

## Figures and Tables

**Figure 1 jcm-11-05591-f001:**
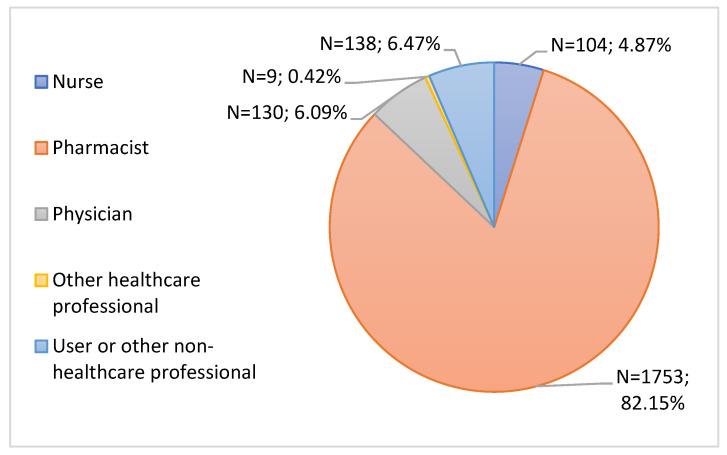
Characterization of notifications by type of notifier.

**Figure 2 jcm-11-05591-f002:**
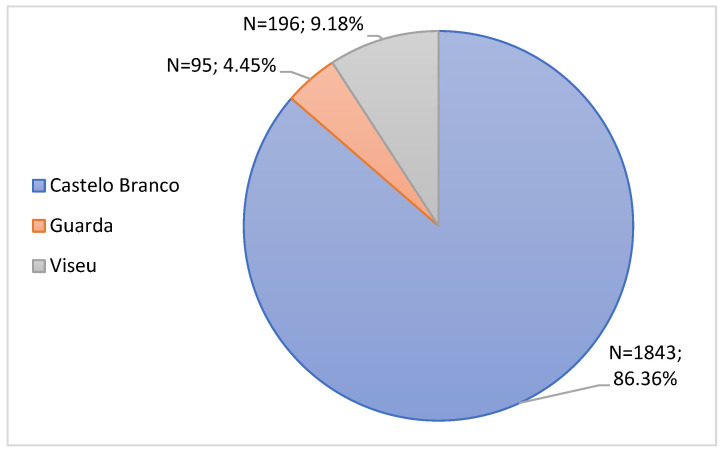
Characterization of notifications by District of Origin of notification.

**Figure 3 jcm-11-05591-f003:**
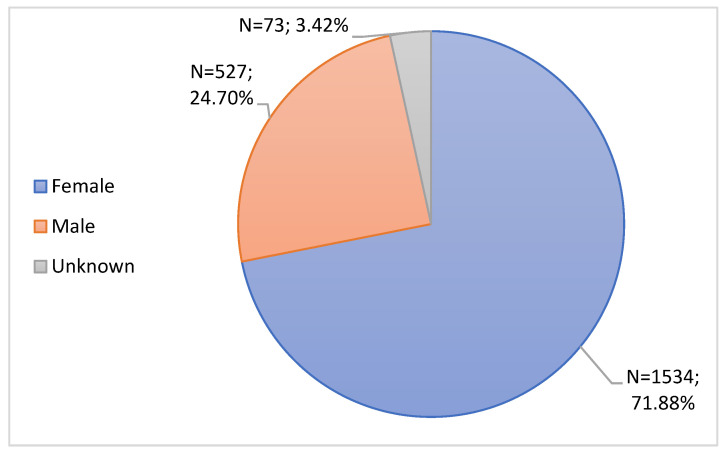
Characterization of notifications according to Gender.

**Figure 4 jcm-11-05591-f004:**
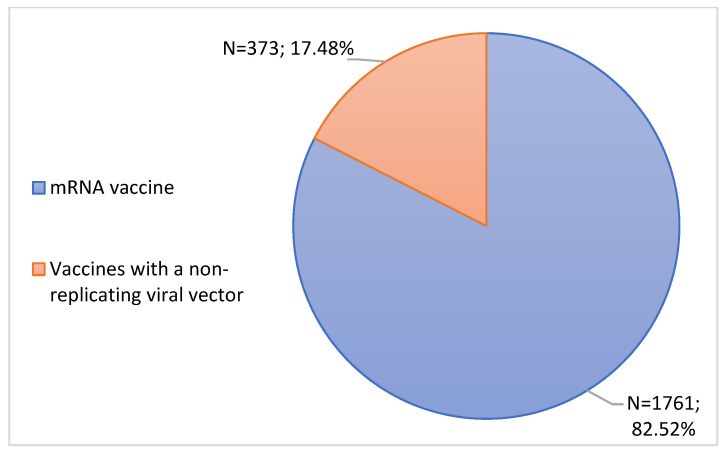
Characterization of notifications according to the type of vaccine.

**Figure 5 jcm-11-05591-f005:**
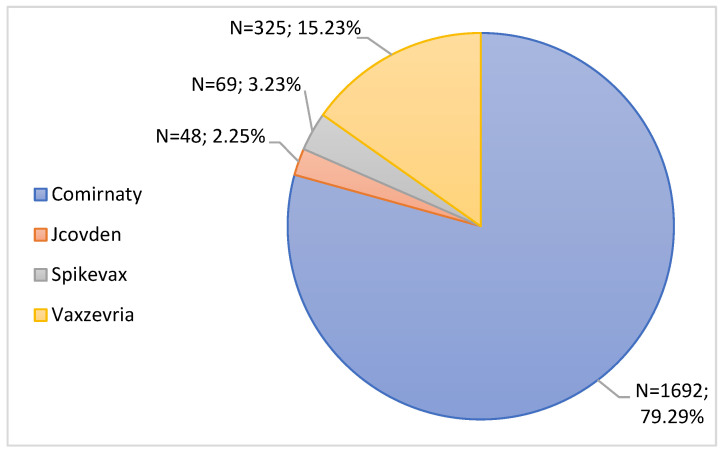
Characterization of notifications according to the brand name of the vaccine.

**Figure 6 jcm-11-05591-f006:**
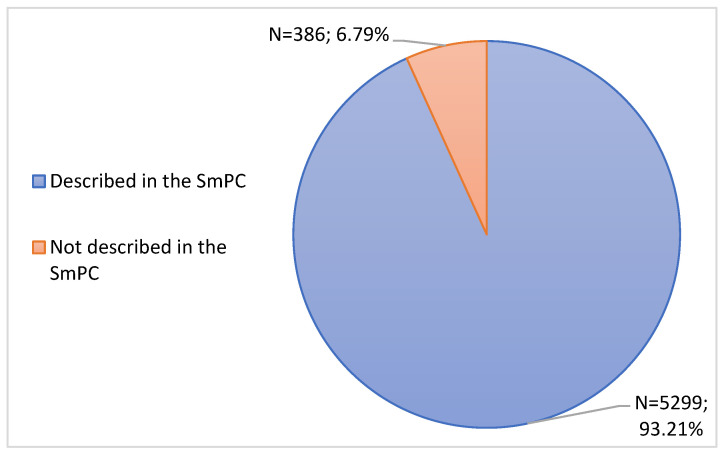
Characterization of Adverse Drug Reactions according to the description in the SmPC.

**Figure 7 jcm-11-05591-f007:**
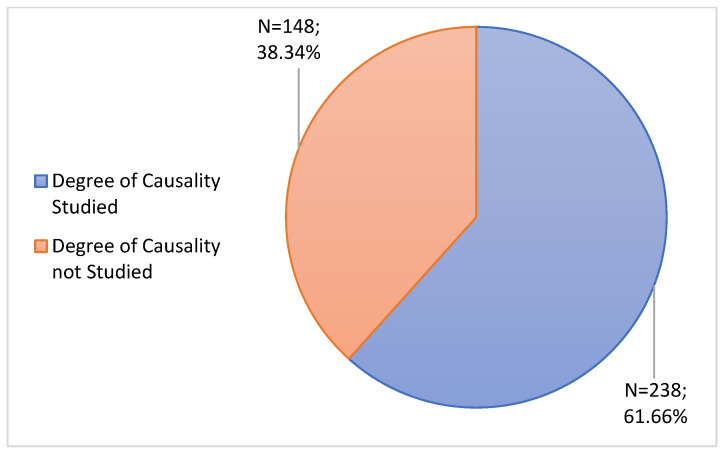
Study of Causality.

**Figure 8 jcm-11-05591-f008:**
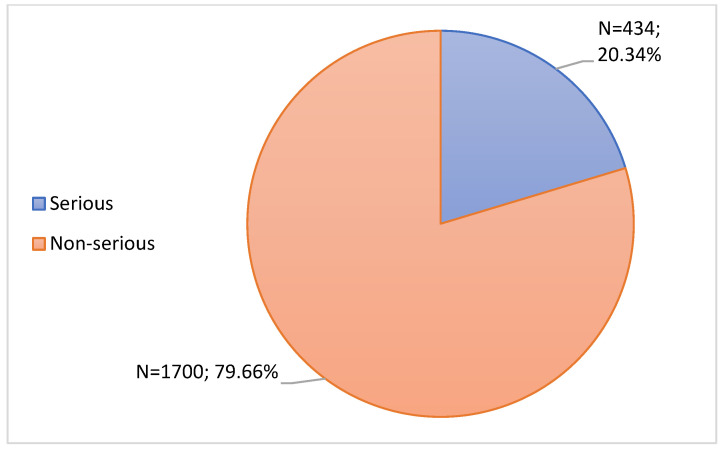
Characterization of notifications according to seriousness.

**Figure 9 jcm-11-05591-f009:**
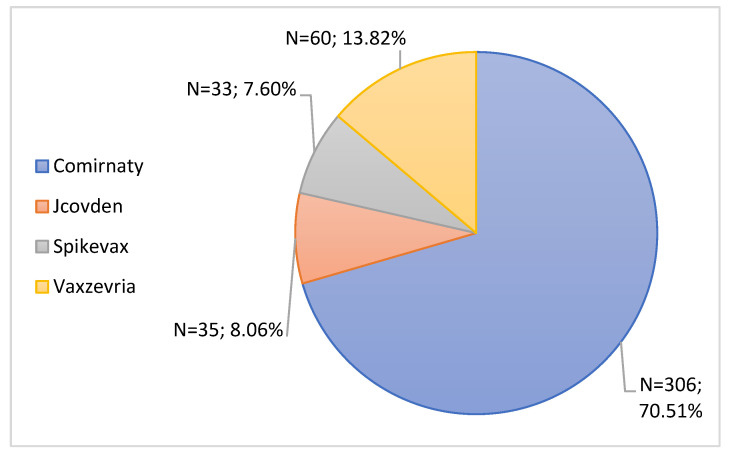
Characterization of serious notifications according to the brand name of the vaccine administered.

**Figure 10 jcm-11-05591-f010:**
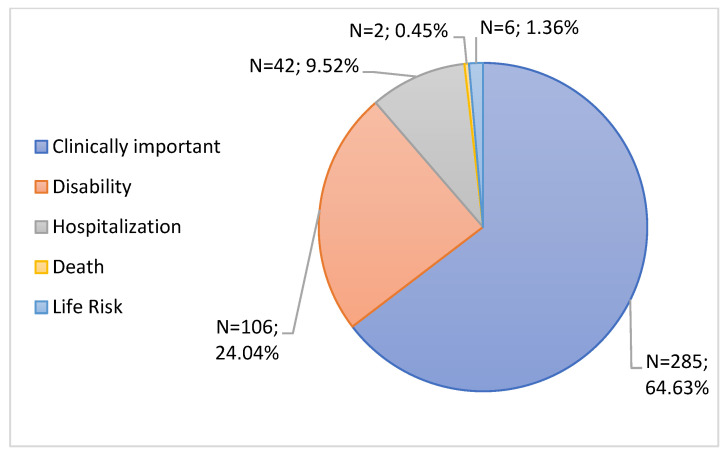
Characterization of serious notifications according to the seriousness criteria.

**Figure 11 jcm-11-05591-f011:**
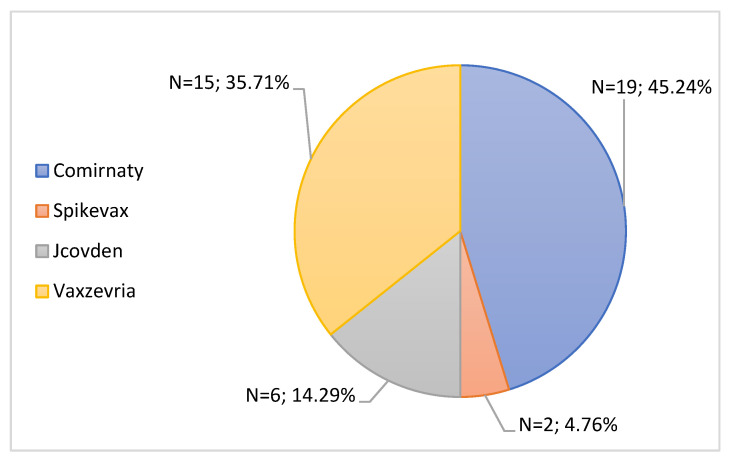
Characterization of serious notifications with the seriousness criteria “Hospitalization”, according to the brand of vaccine administered.

**Figure 12 jcm-11-05591-f012:**
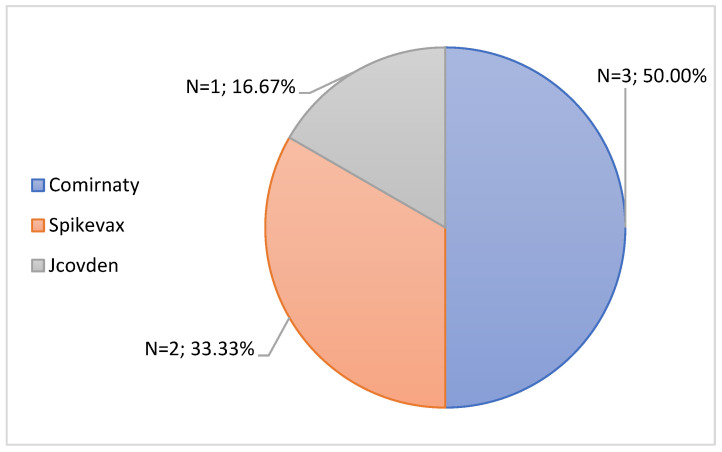
Characterization of serious notifications with the seriousness criteria “Life Risk”, according to the brand of vaccine administered.

**Figure 13 jcm-11-05591-f013:**
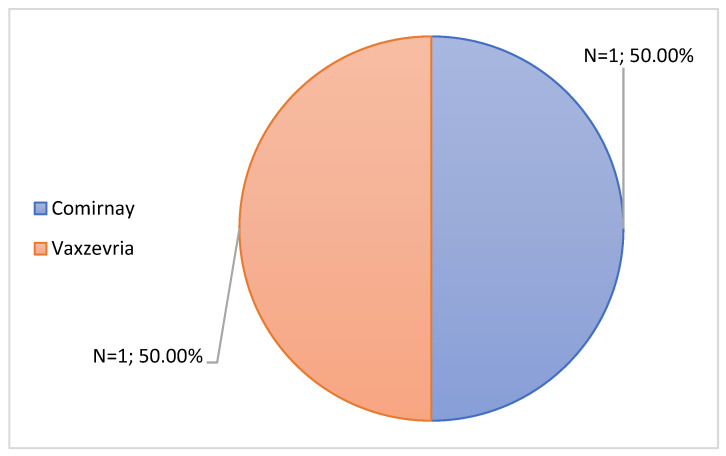
Characterization of serious notifications with the seriousness criteria “Death”, according to the brand of vaccine administered.

**Table 1 jcm-11-05591-t001:** Distribution of notifications by age group.

Age Groups	Frequency	Percentage (%)
(5–11)	0	0.0
(12–17)	7	0.33
(18–24)	108	5.06
(25–49)	1230	57.64
(50–64)	574	26.90
(65–79)	97	4.55
≥80	66	3.09
Unknown	52	2.44
Total	2134	100.00

**Table 2 jcm-11-05591-t002:** Characterization of Adverse Drug Reactions by System Organ Class groups.

System Organ Class Groups	Frequency	Percentage (%)
General disorders and administration site conditions	2454	43.17
Nervous system disorders	1048	18.43
Musculoskeletal and connective tissue disorders	1015	17.85
Gastrointestinal disorders	464	8.16
Blood and lymphatic system disorders	147	2.59
Skin and subcutaneous tissue disorders	144	2.53
Respiratory, thoracic and mediastinal disorders	115	2.02
Infections and infestations	65	1.14
Vascular disorders	62	1.09
Cardiac disorders	42	0.74
Psychiatric disorders	22	0.39
Injury, poisoning and procedural complications	21	0.37
Ear and labyrinth disorders	19	0.33
Eye disorders	16	0.28
Metabolism and nutrition disorders	13	0.23
Investigations	11	0.19
Reproductive system and breast disorders	10	0.18
Immune system disorders	8	0.14
Renal and urinary disorders	6	0.11
Social Circumstances	2	0.04
Pregnancy, puerperium and perinatal conditions	1	0.02
Total	5685	100.00

**Table 3 jcm-11-05591-t003:** Characterization of Adverse Drug Reactions according to the Preferred Term.

Adverse Drug Reaction	Frequency	Percentage (%)
Reaction at or around the site of administration	1147	20.18
Myalgia	751	13.21
Headache	608	10.69
Pyrexia	499	8.78
Chills	222	3.91
Nauseas	219	3.85
Fatigue	214	3.76
Somnolence	178	3.13
Arthralgia	160	2.81
General Pain and Malaise	156	2.74
Lymphadenopathy	141	2.48
Asthenia	106	1.86
Diarrhoea	98	1.72
Dizziness	92	1.62
Vomiting	91	1.60
Pain in extremity	68	1.20
Rash	65	1.14
Change in body temperature	65	1.14
Influenza	63	1.11
Other reactions *	742	13.05
Total	5685	100.00

* The “Other reactions” category encompasses adverse reactions with a notification rate ≤1%.

**Table 4 jcm-11-05591-t004:** Characterization of Adverse Drug Reactions according to the degree of causality attributed.

Causality	Frequency	Percentage (%)
Definitive	0	0.00
Probable	100	42.02
Possible	118	49.58
Unlikely	18	7.56
Conditional	0	0.00
Unclassifiable	2	0.84
Total	238	100.00

**Table 5 jcm-11-05591-t005:** Characterization of serious notifications with the seriousness criteria “Hospitalization”, according to the associated age.

Age Group	Frequency	Percentage (%)
(5–11)	0	0.00
(12–17)	0	0.00
(18–24)	2	4.76
(25–49)	13	30.95
(50–64)	7	16.67
(65–79)	12	28.57
≥80	7	16.67
Unknown	1	2.38
Total	42	100.00

**Table 6 jcm-11-05591-t006:** Characterization of serious notifications with the seriousness criterion “Life Risk”, according to the associated age.

Age Group	Frequency	Percentage (%)
(5–11)	0	0.00
(12–17)	0	0.00
(18–24)	1	16.67
(25–49)	3	50.00
(50–64)	1	16.67
(65–79)	0	0.00
≥80	1	16.67
Unknown	0	0.00
Total	6	100.00

**Table 7 jcm-11-05591-t007:** Evolution of Adverse Drug Reactions associated with vaccines used in immunization against COVID-19.

Evolution of Adverse Drug Reactions	Frequency	Percentage (%)
Cure	5361	94.30
Cure with collateral damage	9	0.16
In recovery	6	0.11
Death	6	0.11
Unknown	303	5.33
Total	5685	100.00

**Table 8 jcm-11-05591-t008:** Relationship between the Adverse Drug Reactions of the notifications that progressed to death with the terms belonging to the Import Medical Event (IME) list.

System Organ Class Group	IME List Terms
Cardiac disorders (3)	Acute myocardial infarction (2); Cardiogenic shock (1)
Renal and urinary disorders (1)	Acute kidney injury (1)
Metabolism and nutrition disorders (1)	Hyperkalaemia (1)

## Data Availability

The raw data used in this research are available to the authors, depending on INFARMED’s authorization.

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
