# Peer review of "COVID-19 Vaccines Adverse Reactions Reported to the Pharmacovigilance Unit of Beira Interior in Portugal"

_jcm, 2022, doi:10.3390/jcm11195591_

Round 1

Reviewer 1 Report

I enjoyed reading this paper and it is an interesting paper on Coronavirus disease-2019 (COVID-19) vaccines and adverse drug reactions (ADRs). However, I have some comments regarding the manuscript. The current study could be strengthened by addressing the following

Abstract

1.      Please use full for COVID-19 and SARS-CoV-2.

2.      Please also mention the exact date “December 2020 and December 2021”

3.      Try to use the frequency of cases along with percentage e.g., 20.34% (n=?)

4.      Please correct these typographic mistakes “9,52% and 0,45%” and try to mention them as 9.52%....

5.      Please avoid the use of percentages at the start of a sentence. “20.34% of the notifications and 94.30% of ADRs resolved” these two sentences should be rearranged.

Introduction

1.      Please use full for COVID-19 and SARS-CoV-2 when explaining the first time.

2.      If possible cite these articles in the Introduction section https://doi.org/10.1007/s40267-021-00852-z, https://doi.org/10.1016/j.phclin.2021.06.001, and https://doi.org/10.3396/ijic.v17.21250”.

3.      Please remove “between December 2020 and December 2021” from the objective sentence because this is already highlighted in the abstract as well as in the material and method section.

Material and Method

1.      Please mention the exact date “December 2020 and December 2021”

2.      Please mention the company name and country of manufacturing for “Microsoft Office 98 Excel 365 tool”

3.      The authors used different criteria for ADRs causality assessment and gives three references “Sistemas de Imputação e Avaliação da Causalidade” “Comparison of two methods to assess causality of adverse drug reactions” and “The use of the WHO-UMC system for standardized case causality assessment”. Why do you use these three criteria please give a reason.

4.      Similarly, why do you use “Good pharmacovigilance practices. European Medicines Agency” to evaluate “Seriousness and Seriousness Criteria”? Give reason.

Results

1.      How do you assess the causality of hepatic ADRs? Via WHO-UMC criteria or any other?

2.      Please replace “Sequelae” with another suitable word for a clear understanding of the general reader.

3.      If possible discuss the management of ADRs. Means what were the actions adopted by HCPs in response to ADRs (vaccination stopped/continue/ lower the dose etc.)

Limitations

1.      The manuscript has more limitations. Please add and elaborate.

Strengths of study

1.      Please also include the strengths of the study

2.      Include what is the results brought to practical clinics

Conclusion

1.      The conclusion is not clear (too long). Just focus on the objectives (to characterize the ADRs associated with vaccines used in immunization against COVID-19 and results obtained with the safety data from studies carried out in other countries around the world)

2.      Remove the first paragraph from the conclusion

References

1.      Update the references and include the suggested ones

2.      Please also add English translation inside the bracket for 14 to 17, 19, 30, and 41 references for a clear understanding of international readers.

Author Response

I enjoyed reading this paper and it is an interesting paper on Coronavirus disease-2019 (COVID-19) vaccines and adverse drug reactions (ADRs). However, I have some comments regarding the manuscript. The current study could be strengthened by addressing the following

Author´s reply: Thank you very much for your comments and for showing interest in our work. As requested, the manuscript was revised according to the comments and suggestions received, which were important to improve the quality of this manuscript.

Abstract

  1. Please use full for COVID-19 and SARS-CoV-2.

Authors´ reply: Thank you very much for your suggestion, which was taken into consideration. We revised the abstract and add the definition of the words COVID-19 and SARS-CoV-2 in the abstract (please see the Revised Manuscript).

  1. Please also mention the exact date “December 2020 and December 2021”

Authors´ reply: Thank you very much for your suggestion, which was taken into consideration. We fully agree with the Reviewer. In fact, by our mistake, we did not correctly mention the period during which this study took place. We have already changed it to “December 1st 2020 and December 31st 2021” (please see the Revised Manuscript).

  1. Try to use the frequency of cases along with percentage e.g., 20.34% (n=?)

Authors´ reply: Thank you very much for your suggestion, which was taken into consideration. We revised the abstract and add the frequency of cases (please see the Revised Manuscript).

  1. Please correct these typographic mistakes “9,52% and 0,45%” and try to mention them as 9.52%....

Authors´ reply: Thank you for your suggestion. In fact, by our mistake, these 2 data did not have the correct configuration, which we have already changed (please see the Revised Manuscript).

  1. Please avoid the use of percentages at the start of a sentence. “20.34% of the notifications and 94.30% of ADRs resolved” these two sentences should be rearranged.

Authors´ reply: Thank you for your suggestion. In fact, these 2 sentences were not written in the most correct way which we have already changed (please see the Revised Manuscript).

Introduction

  1. Please use full for COVID-19 and SARS-CoV-2 when explaining the first time.

Authors´ reply: Thank you very much for your suggestion, which was taken into consideration. We revised the introduction and add the definition of the words COVID-19 and SARS-CoV-2, as can be seen in lines 29 and 30. (please see the Revised Manuscript).

  1. If possible cite these articles in the Introduction section https://doi.org/10.1007/s40267-021-00852-z, https://doi.org/10.1016/j.phclin.2021.06.001, and https://doi.org/10.3396/ijic.v17.21250”.

Authors´ reply: Thank you very much for your suggestion. We revised the introduction and we have cited all the articles referred  in line 83, which is related to the topic of this article addressing the importance of pharmacovigilance (please see the Revised Manuscript).

  1. Please remove “between December 2020 and December 2021” from the objective sentence because this is already highlighted in the abstract as well as in the material and method section.

Authors´ reply: Thank you very much for your suggestion, which was taken into consideration. We revised the introduction and removed “between December 2020 and December 2021” (please see the Revised Manuscript).

Material and Method

  1. Please mentionthe exact date “December 2020 and December 2021”

Authors´ reply: Thank you very much for your suggestion, which was taken into consideration. We fully agree with the Reviewer. We have already changed it to “December 1st 2020 and December 31st 2021”, as can be seen in line 98 (please see the Revised Manuscript).

  1. Please mention the company name and country of manufacturing for “Microsoft Office 98 Excel 365 tool”

Authors´ reply: The tool used in this study, Microsoft Office 98 Excel 365 tool, belongs to Microsoft Corporation, a US based company.

  1. The authors used different criteria for ADRs causality assessment and gives three references “Sistemas de Imputação e Avaliação da Causalidade” “Comparison of two methods to assess causality of adverse drug reactions” and “The use of the WHO-UMC system for standardized case causality assessment”. Why do you use these three criteria please give a reason.

Authors´ reply: In fact, by our mistake, we cited 3 articles that talk about various methods of attributing causality categories. However, we did not state that we intended to follow the attribution of causality categories according to the WHO. Thus, we decided to leave only one citation that refers to the definition of the degrees of causality according to the WHO, as can be seen in line 163 (please see the Revised Manuscript).

  1. Similarly, why do you use “Good pharmacovigilance practices. European Medicines Agency” to evaluate “Seriousness and Seriousness Criteria”? Give reason.

Authors´ reply: We cite this reference because it is related to the definition of serious reaction present in lines 171-173 of this manuscript. This definition can be found in the document - Guideline on good pharmacovigilance practices (GVP) - Module VI - Collection, management and submission of reports of suspected adverse reactions to medicinal products - on page 9. We have changed the reference to make it clearer which document we are referring to, as can be seen in line 608 (please see the Revised Manuscript).

Results

  1. How do you assess the causality of hepatic ADRs? Via WHO-UMC criteria or any other?

Authors´ reply: The degree of causality is assigned to each ADR by experts who perform the assessment according to WHO criteria and based on the information provided in the notification, taking into account the ADR, the suspected drug and clinical information provided by the notifier.

  1. Please replace “Sequelae” with another suitable word for a clear understanding of the general reader.

Authors´ reply: Thank you very much for your suggestion, which was taken into consideration. We have already changed it to “Cure with collateral damage”, as can be seen in lines 356 and 357 (please see the Revised Manuscript).

  1. If possible discuss the management of ADRs. Means what were the actions adopted by HCPs in response to ADRs (vaccination stopped/continue/ lower the dose etc.)

Authors´ reply: Thank you very much for your suggestion. In fact, during the study period, several countries discontinued administration of some vaccines or restricted their use to certain populations due to the appearance of rare and serious adverse reactions. So we added a phrase to the discussion that portrays this, as can be seen in lines 482-484 (please see the Revised Manuscript). However, the suspension of the vaccine does not apply since the vaccines are for a single administration, so the patient cannot suspend the administration of the drug. On the other hand, following some serious ADRs, patients may not have taken booster doses, but it is not possible to obtain this information through notifications.

Limitations

  1. The manuscript has more limitations. Please add and elaborate.

Authors´ reply: Thank you very much for your suggestion, which was taken into consideration. In fact, we found another limitation that may be associated with our study, as can be seen in lines 535-538. The fact that the Comirnaty vaccine is the most administered vaccine in Portugal, may have biased the results obtained, since the vaccines under study were not administered in the same number of patients (please see the Revised Manuscript).

Strengths of study

  1. Please also include the strengths of the study

  1. Include what is the results brought to practical clinics

Authors´ reply: Thank you very much for your suggestion, which was taken into consideration. We add a section with the strengths of our study (please see the Revised Manuscript).

Conclusion

  1. The conclusion is not clear (too long). Just focus on the objectives (to characterize the ADRs associated with vaccines used in immunization against COVID-19 and results obtained with the safety data from studies carried out in other countries around the world)

  1. Remove the first paragraph from the conclusion

Authors´ reply: Thank you very much for your suggestion, which was taken into consideration. We revised and updated the conclusion (please see the Revised Manuscript).

References

  1. Update the references and include the suggested ones

  1. Please also add English translation inside the bracket for 14 to 17, 19, 30, and 41 references for a clear understanding of international readers.

Authors´ reply: Thank you very much for your suggestion, which was taken into consideration. We revised and updated the references (please see the Revised Manuscript).

Reviewer 2 Report

The aim of this study is to characterize the adverse drug reactions (ADRs) of COVID-19 vaccines between December 2020 and December 2021. 16 2134 notifications were studied corresponding to 5685 ADRs. 20.34% of the notifications were considered serious reactions, of which 9.52% resulted in hospitalization and 0.45% resulted in death. Authors revealed that among the ADRs notified, reactions at or around the injection site, myalgias, headache and pyrexia. 

The paper is well written and properly planned. The results are reliable because they involve a very large group of patients.

What I find missing from the discussion are a few sentences on headache as an adverse effect including headache associated with serious side effects like thrombosis. The most recent meta-analysis on this topic was published by Castaldo et al. and is worth citing and using in the discussion. https://pubmed.ncbi.nlm.nih.gov/35361131/

Author Response

The aim of this study is to characterize the adverse drug reactions (ADRs) of COVID-19 vaccines between December 2020 and December 2021. 16 2134 notifications were studied corresponding to 5685 ADRs. 20.34% of the notifications were considered serious reactions, of which 9.52% resulted in hospitalization and 0.45% resulted in death. Authors revealed that among the ADRs notified, reactions at or around the injection site, myalgias, headache and pyrexia. 

The paper is well written and properly planned. The results are reliable because they involve a very large group of patients.

What I find missing from the discussion are a few sentences on headache as an adverse effect including headache associated with serious side effects like thrombosis. The most recent meta-analysis on this topic was published by Castaldo et al. and is worth citing and using in the discussion. https://pubmed.ncbi.nlm.nih.gov/35361131/

Authors´ reply: Thank you very much for your comments and for showing interest in our work. We have already added an explanation of why headaches are among the most notified ADRs, as can be seen in lines 463 - 472 (please see the Revised Manuscript).